# Reducing the Kidney Uptake of High Contrast CXCR4 PET Imaging Agents via Linker Modifications

**DOI:** 10.3390/pharmaceutics14071502

**Published:** 2022-07-20

**Authors:** Daniel Kwon, Zhengxing Zhang, Jutta Zeisler, Hsiou-Ting Kuo, Kuo-Shyan Lin, Francois Benard

**Affiliations:** 1Department of Molecular Oncology, BC Cancer, Vancouver, BC V5Z 1L3, Canada; dkwon@bccrc.ca (D.K.); zzhang@bccrc.ca (Z.Z.); jzeisler@bccrc.ca (J.Z.); htkuo0325@gmail.com (H.-T.K.); klin@bccrc.ca (K.-S.L.); 2Department of Radiology, University of British Columbia, Vancouver, BC V5Z 1M9, Canada

**Keywords:** positron emission tomography, oncology, CXCR4, pharmacokinetics, nuclear medicine

## Abstract

Purpose: The C-X-C chemokine receptor 4 (CXCR4) is highly expressed in many subtypes of cancers, notably in several kidney-based malignancies. We synthesized, labeled, and assessed a series of radiotracers based on a previous high contrast PET imaging radiopharmaceutical [^68^Ga]Ga-BL02, with modifications to its linker and metal chelator, in order to improve its tumor-to-kidney contrast ratio. Methods: Based on the design of BL02, a piperidine-based cationic linker (BL06) and several anionic linkers (tri-Aad (BL17); tri-D-Glu (BL20); tri-Asp (BL25); and tri-cysteic acid (BL31)) were substituted for the triglutamate linker. Additionally, the DOTA chelator was swapped for a DOTAGA chelator (BL30). Each radiotracer was labeled with ^68^Ga and evaluated in CXCR4-expressing Daudi xenograft mice with biodistribution and/or PET imaging studies. Results: Of all the evaluated radiotracers, [^68^Ga]Ga-BL31 showed the most promising biodistribution profile, with a lower kidney uptake compared to [^68^Ga]Ga-BL02, while retaining the high imaging contrast capabilities of [^68^Ga]Ga-BL02. [^68^Ga]Ga-BL31 also compared favorably to [^68^Ga]Ga-Pentixafor, with superior imaging contrast in all non-target organs. The other anionic linker-based radiotracers showed either equivocal or worse contrast ratios compared to [^68^Ga]Ga-BL02; however, [^68^Ga]Ga-BL25 also showed lower kidney uptake, as compared to that of [^68^Ga]Ga-BL02. Meanwhile, [^68^Ga]Ga-BL06 had high non-target organ uptake and relatively lower tumor uptake, while [^68^Ga]Ga-BL30 showed significantly increased kidney uptake and similar tumor uptake values. Conclusions: [^68^Ga]Ga-BL31 is an optimized CXCR4-targeting radiopharmaceutical with lower kidney retention that has clinical potential for PET imaging and radioligand therapy.

## 1. Introduction

To this day, the rational modification of pharmacophores to enhance binding potency and selectivity for lead optimization forms the backbone of drug development strategies [1,2,3,4]. An equally important consideration for lead optimization, however, is the rational alterations to optimize a pharmacophore’s pharmacokinetic performance, such as bioavailability and metabolism, that impact the ligand’s potential as a lead [5,6,7]. Enhancing properties such as resistance to enzymatic degradation [8] or circulation residence time [9] is essential for the success of a lead ligand as a potential therapeutic.

Radiopharmaceuticals for imaging and therapy are highly reliant on their pharmacokinetic properties for maximal imaging contrast and therapeutic index [10]. Radiotracers for imaging purposes (e.g., single photon emission computed tomography (SPECT) or positron emission tomography (PET)) must not only bind selectively to the desired target at high affinity, but also show a comparatively lower uptake in organs and tissues that do not express the desired target. The difference in measured radioactivity between the diseased tissue and surrounding area is vital for maximal imaging contrast. Similarly, enhancing the uptake of a radiotracer in target lesions while reducing uptake in healthy tissue reduces off-target radiotoxicity, while maximizing therapeutic outcomes. Determinants of high contrast PET or SPECT images include stability, hydrophilicity, affinity to the biological target and retention at the pathological site of the administered radiopharmaceutical. As modifying functional groups on the pharmacophore for pharmacokinetic optimization may ablate its affinity to the target, strategies for radiopharmaceutical optimization for imaging firstly focus on linker and radioprosthesis modifications [11,12,13,14].

The C-X-C chemokine receptor 4 (CXCR4) is a chemokine receptor that is highly expressed in over 20 subtypes of cancers [15,16,17]. It contributes to activating downstream proliferative pathways, enabling resistance to chemotherapeutics and enhancing the metastatic potential of cancer cells [15,17,18,19,20,21]. Targeting CXCR4 with radiopharmaceuticals has emerged as a viable method of identifying and potentially treating patients with CXCR4-expressing cancers, with [^68^Ga]Ga-Pentixafor and [^177^Lu]Lu-/[^90^Y]Y-Pentixather leading the way in clinical studies [22,23]. 

We previously reported a series of radiopharmaceuticals targeting CXCR4 by using the LY2510924 antagonist as the targeting pharmacophore. [^68^Ga]Ga-/[^177^Lu]Lu-BL02, possessing a triglutamate-based linker, showed excellent PET imaging properties [24]. The purpose of this study was to further optimize BL02, notably with the aim of improving the tumour-to-kidney contrast ratios for the eventual development of a CXCR4 therapeutic radioligand.

Kidney uptake is primarily mediated by resorption of the radioligand or radiometabolites through receptor- or transporter-mediated processes such as megalin [25,26,27]. Many of these interactions involve charged elements that are present in the linker and/or metal–chelator complex. Furthermore, previous SAR studies have revealed the impact of linkers on this particular line of radiopharmaceuticals. For example, the use of an amide-based linker (BL08) over a triazole (BL09) resulted in significantly lower kidney retention [28]. To better understand the structural determinants of the pharmacokinetics of our CXCR4-targeting radiopharmaceuticals, we altered the triglutamate linker with a cationic piperidine-based linker. This was motivated by our previous work on the development of MC1R-targeting radiopharmaceuticals, which showed that a piperidine-based linker was essential in increasing the tumor-to-kidney ratios [29]. Furthermore, given the highly desirable properties of triglutamate-based linkers in this context, we hypothesized that radiotracers with linkers or chelators possessing similar physiochemical properties to that of BL02 may reduce kidney uptake, while retaining similar tumor uptake and overall pharmacokinetic properties.

## 2. Materials and Methods

### 2.1. Chemistry

All reagents and solvents were purchased from commercial sources and used without further purification. All peptides were synthesized on a Liberty Blue automated microwave peptide synthesizer (CEM Corporations, Matthews, NC, USA), unless otherwise indicated. High performance liquid chromatography (HPLC) was performed on an (1) Agilent 1260 Infinity system equipped with a model 1200 quaternary pump, a model 1200 UV absorbance detector (set at 220 nm), and a Bioscan NaI scintillation detector; (2) Agilent 1260 Infinity II preparative system equipped with a model 1260 Infinity II preparative binary pump, a model 1260 Infinity variable wavelength detector (set at 220 nm), and a 1290 Infinity II preparative open-bed fraction collector. The HPLC columns used were a semi-prep column (C18, 5 μm, 250 mm × 10 mm); an analytical column (C18, 5 μm, 250 mm × 4.6 mm); and a preparative column (Gemini, NX-C18, 5 μm, 50 mm × 30 mm), all purchased from Phenomenex. Mass analyses were performed using an AB SCIEX 4000 QTRAP mass spectrometer system with an ESI ion source.

### 2.2. Chemical Synthesis

Information on the synthesis of BL06, BL17, BL20, BL25, BL30 and BL31 are described in the Appendix A.

### 2.3. Radiochemistry

To label CXCR4-targeting DOTA-conjugated peptide precursors with ^68^Ga, [^68^Ga]GaCl_3_ was eluted from an ITG generator (ITM, Munich, Germany) with a total of 4 mL of 0.05 M HCl. The eluted [^68^Ga]GaCl_3_ solution was added to 2 mL of concentrated HCl. This radioactive mixture was then added to a DGA resin column and washed with 3 mL of 5 M HCl. The column was then dried with air and the [^68^Ga]GaCl_3_ (0.10–0.50 GBq) was eluted with 0.5 mL of water. This solution was added to a solution of the precursor (25 μg) in 0.7 mL HEPES buffer (2 M, pH 5.3). The reaction mixture was heated in a microwave oven (Danby; DMW7700WDB) for 1 min at power setting 2. The mixture was purified by HPLC using a semi-prep column. Quality control was performed on a HPLC system equipped with an analytical column. Molar activities were calculated based on a standard curve that was generated from injections of cold standards on the HPLC with a semi-prep column at a wavelength of 220 nm, except for that of Pentixafor, which was calculated using a standard curve that was generated from injections of the precursor on the HPLC with a semi-prep column at a wavelength of 269 nm.

### 2.4. Cell Culture

The Daudi B lymphoblast cell line was purchased from the American Type Culture Collection (ATCC CCL-213). The CHO:CXCR4 cell line was a kind gift from Drs. David McDermott and Xiaoyuan Chen (National Institutes of Health, Bethesda, MD, USA). The Daudi and CHO:CXCR4 cell lines were cultured in a 5% CO_2_ atmosphere at 37 °C in a humidified incubator with RPMI-1640 medium (Life Technologies Corporations, Carlsbad, CA, USA). Culture media were supplemented with 10% fetal bovine serum (Sigma-Aldrich, St. Louis, MI, USA), 100 IU/mL penicillin, and 100 μg/mL streptomycin (penicillin–streptomycin solution). All cell lines were verified for murine pathogens and mycoplasma contamination by the Impact 1 mouse profile (Idexx BioAnalytics, Columbia, MO, USA).

### 2.5. In Vitro Competitive Binding Assay

The binding affinities of non-radioactive Ga-conjugated peptides were determined using a cell-based competition binding assay. CHO:CXCR4 cells were seeded at a density of 1 × 10^5^ cells per well in 24-well poly-D-lysine coated plates (Corning BioCoat) and incubated with [^125^I]SDF-1α (0.01 nM, PerkinElmer, Waltham, MA, USA) and competing nonradioactive ligands (1 μM to 0.1 pM). The cells, radioligand, and competing peptides were incubated for 1 h at 27 °C with moderate shaking. Afterwards, the supernatant was aspirated, and the wells were washed with 1 mL of ice-cold PBS. The cells were harvested with 200 μL of trypsin and radioactivity was measured on the Hidex AMG Automatic Gamma Counter. Data were plotted in GraphPad Prism 7 to determine the IC_50_ values (GraphPad Software, Inc., La Jolla, CA, USA). The values are reported as mean ± standard deviation.

### 2.6. LogD_7.4_ Measurement

[^68^Ga]Ga-BL31 was aliquoted into vials with 3 mL of octanol and 3 mL of 0.1 M phosphate buffer (pH 7.4). Each vial was vortexed (1 min) and centrifuged (5000 RPM, 10 min). The octanol and aqueous layers were sampled (1 mL) and counted in a well counter. LogD_7.4_ was calculated using the following equation: LogD_7.4_ = log_10_[(counts in octanol phase)/(counts in buffer phase)].

### 2.7. Animal Models

The animal experiments were performed in accordance with the guidelines established by the Canadian Council on Animal Care and approved by the Animal Ethics Committee of the University of British Columbia. Male NOD.Cg-Rag1tm1Mom Il2rgtm1Wjl/SzJ (NRG) mice were obtained from an in-house breeding colony at the Animal Resource Centre of the BC Cancer Research Centre, Vancouver, Canada. Mice (6–10 weeks old, 25–30 g weight) were subcutaneously inoculated with 5 × 10^6^ Daudi cells (100 μL; 1:1 ratio of PBS/Matrigel) on the left flank. Tumors were grown for 18–23 days before in vivo experiments.

### 2.8. PET/CT Imaging

PET and CT scans were performed on a Siemens Inveon microPET/CT. The tumor-bearing mice were briefly anesthetized with isoflurane (2–2.5% isoflurane in 2 L/min O_2_) for an i.v. injection of 4–7 MBq (10–700 pmol) of the ^68^Ga-labelled peptide. The mice received an intraperitoneal (i.p.) injection of 7.5 µg (0.25–0.3 mg/kg) LY2510924 15 min prior to radiotracer administration as blocking controls. The animals were allowed to roam freely during the uptake period (50 or 110 min), after which they were anesthetized and scanned. Xenograft mice were not imaged longitudinally. The CT scan was obtained for attenuation correction and anatomical localization (80 kV; 500 μA; 3 bed positions; 34% overlap; 220° continuous rotation), followed by a 10 or 15 min PET acquisition at 1 or 2 h p.i. of the radiotracer. The PET data were acquired in list mode, reconstructed using 3-dimensional ordered-subsets expectation maximization (2 iterations), followed by a fast maximum a priori algorithm (18 iterations) with CT-based attenuation correction. Images were analyzed using the Inveon Research Workplace software (Siemens Healthineers, Malern, PA, USA).

### 2.9. Biodistribution Studies

Under isoflurane anesthesia (2–2.5% isoflurane in 2 L/min O_2_), the mice were injected intravenously with 0.8–3.0 MBq of the ^68^Ga-labelled peptide and euthanized at the selected timepoints. Additional groups of mice received 7.5 µg (0.25–0.3 mg/kg) LY2510924 as a blocking control i.p. 15 min before radiotracer injection, and euthanized 1 h p.i. Tissues were harvested, washed in PBS, blotted dry, and weighed, and radioactivity was counted on a Hidex AMG Automatic Gamma Counter. The counts were decay corrected, converted to absolute units using a calibration curve, and expressed as percent injected dose per gram of tissue (%ID/g).

### 2.10. Statistical Analysis

The statistical analyses were performed by GraphPad Prism 8 and R (R Foundation for Statistical Computing, Great Lakes, MI, USA, version 4.2.0.). Continuous variables are reported as mean ± standard deviation. The IC_50_ values were compared between groups using ANOVA that was corrected for multiple comparisons with Dunnett’s method. Tissue uptake was compared between blocked and unblocked groups at 1 h using Welch’s *t*-test that was corrected for multiple comparisons. Radiotracers were compared using either Welch’s *t*-test or ANOVA that were corrected for multiple comparisons with Dunnett’s method, with a post-hoc Tukey test. The threshold for significance was set at *p* < 0.05. The ROUT method was used to identify outliers (α = 0.01) [30].

## 3. Results

### 3.1. Synthesis of the Derivatives

To assess the effect of a cationic linker, we elected to use a piperidine functional group (Pip) as a linker over the triglutamate construct (BL06). Furthermore, to retain the anionic charge, the linker was modified to a tri-homoglutamate (Aad, BL17); tri-D-glutamate (D-Glu, BL20); tri-aspartate (Asp, BL25); and tri-cysteic acid (CysAcid, BL31). Finally, to assess the effect of a chelator with a greater hydrophilic property without affecting the linker length, we opted to conjugate a DOTAGA chelator over the DOTA via the tetra-tert-butyl protected DOTAGA (BL30) [31]. Each derivative was conjugated with non-radioactive GaCl3 to form a non-radioactive standard for further in vitro characterization (Figure 1) [32,33].

### 3.2. In Vitro Characterization and Radiolabeling of BL02 Derivatives

All gallium standards of the aforementioned derivatives showed similar in vitro affinity to CXCR4 as compared to Ga-BL02 (except that of Ga-BL17, which showed a slightly increased affinity, though it was not statistically significant (Table 1)). Ga-BL06 was assessed in two separate in vitro binding experiments (*n* = 2), which showed significant variability. All compounds were synthesized with high molar activity, radiochemical purity, and yield. However, [^68^Ga]Ga-BL06 had closer co-elution of the radiolabeled peak and the unlabeled precursor, resulting in reduced molar activity. The parameters of their radiosyntheses can be found in Appendix A.

### 3.3. Biodistribution and PET Imaging Studies

In vivo PET imaging and ex vivo biodistribution studies of [^68^Ga]Ga-BL06 were performed in the same Daudi xenograft mice previously used to evaluate [^68^Ga]Ga-BL02 [24]. [^68^Ga]Ga-BL06 had a higher uptake in Daudi xenografts on biodistribution and PET imaging at 2 h p.i. than [^68^Ga]Ga-BL02 (11.32 ± 1.44 %ID/g, *p* = 0.00015), but showed no significant difference at 1 h p.i. (10.26 ± 1.29 %ID/g, *p* = 0.16) (Figure 2). Uptake was shown to be specific based on blocking controls (Appendix A). The presence of a cationic linker resulted in a slower rate of excretion from circulation as compared to [^68^Ga]Ga-BL02 (2.78 ± 0.46 vs. 0.27 ± 0.06 %ID/g (*p* < 0.0001) and 1.11 ± 0.24 vs. 0.08 ± 0.04 %ID/g (*p* < 0.0001) in the blood at 1 and 2 h p.i., respectively). Furthermore, [^68^Ga]Ga-BL06 had higher uptake than [^68^Ga]Ga-BL02 in the lungs (15.00 ± 2.06 vs. 0.45 ± 0.08 %ID/g (*p* < 0.0001) and 7.18 ± 0.86 vs. 0.27 ± 0.09 %ID/g (*p* < 0.0001) at 1 and 2 h p.i., respectively); liver (8.60 ± 0.77 vs. 0.57 ± 0.07 %ID/g (*p* < 0.0001) and 7.18 ± 0.86 vs. 0.54 ± 0.04 %ID/g (*p* < 0.0001) at 1 and 2 h p.i., respectively); and spleen (15.53 ± 1.83 vs. 0.47 ± 0.22 %ID/g (*p* < 0.0001) and 9.14 ± 1.60 vs. 0.25 ± 0.10 %ID/g (*p* < 0.0001) at 1 and 2 h p.i., respectively). These findings in the biodistribution study correlated well with the findings on PET imaging, wherein the tumor was visualized, but many of the non-target organs were also visualized (Figure 2).

[^68^Ga]Ga-BL20 (D-Glu) showed a similar biodistribution profile compared to [^68^Ga]Ga-BL02 (Figure 3), with relatively high and similar tumor uptake values (9.07 ± 0.76 and 8.01 ± 1.39 %ID/g at 1 and 2 h p.i., respectively) with no statistically significant difference. Similarly, the blood pool uptake of [^68^Ga]Ga-BL20 was low and cleared rapidly (0.42 ± 0.12 and 0.08 ± 0.02 %ID/g at 1 and 2 h p.i., respectively), with a low uptake throughout most of the non-target organs. For the kidneys, [^68^Ga]Ga-BL20 showed no statistically significant difference in uptake compared to [^68^Ga]Ga-BL02 (3.58 ± 0.26 and 3.02 ± 0.22 %ID/g at 1 and 2 h p.i., respectively).

[^68^Ga]Ga-BL17 (Aad) maintained a similar biodistribution profile as compared to [^68^Ga]Ga-BL02, with no differences in kidney uptake (Figure 3). However, the tumor uptake was lower (6.32 ± 0.67 and 6.40 ± 1.60 %ID/g at 1 and 2 h p.i., respectively), which was significantly lower at 1 h p.i. (*p* = 0.0015) but similar at 2 h p.i. (*p* = 0.1992) compared to [^68^Ga]Ga-BL02. PET imaging with [^68^Ga]Ga-BL17 that was performed at 1 h p.i. further confirmed these results (Figure 3A). Further in vivo assessment of this candidate radiotracer was not pursued due to its non-superiority over [^68^Ga]Ga-BL02.

[^68^Ga]Ga-BL25 (Asp) biodistribution data were only collected at 2 h p.i. [^68^Ga]Ga-BL25 demonstrated a similar biodistribution profile compared to the other carboxylate-based amino acid linkers (Figure 3). The retention in blood was slightly higher than [^68^Ga]Ga-BL02 at 2 h p.i. (0.18 ± 0.01 %ID/g, *p* < 0.0001). The uptake in the kidneys was lower than [^68^Ga]Ga-BL02 (1.92 ± 0.07 vs. 3.40 ± 0.51 %ID/g, *p* < 0.0001), but as the tumor uptake was also lower (5.53 ± 0.29 %ID/g, *p* = 0.054), this resulted in lower tumor-to-organ ratios.

The use of DOTAGA over DOTA led to a concurrent increase in uptake in the kidneys. At 1 h p.i., [^68^Ga]Ga-BL30 had an 8.99 ± 1.56 %ID/g uptake in the kidneys, more than twice that of [^68^Ga]Ga-BL02. Furthermore, there was a significant decrease in tumor uptake (5.76 ± 0.77 %ID/g, *p* < 0.0001) There were no other remarkable differences in the biodistribution of [^68^Ga]Ga-BL30 as compared to [^68^Ga]Ga-BL02.

[^68^Ga]Ga-BL31, possessing three cysteic acids within its linker, exhibited a similar biodistribution profile to [^68^Ga]Ga-BL02; however, it had a lower kidney uptake at both 1 and 2 h timepoints (2.54 ± 0.26 vs. 3.81 ± 0.86 %ID/g (*p* = 0.0037) and 2.23 ± 0.21 vs. 3.40 ± 0.51 %ID/g (*p* < 0.0001) at 1 h and 2 h p.i, respectively) (Figure 4). There was no statistically significant difference in tumor uptake (9.41 ± 1.00 %IDg (*p* = 0.70) and 8.94 ± 1.45 %ID/g (*p* = 0.13) at 1 and 2 h p.i, respectively). The specificity of uptake was further confirmed with blocking studies (0.94 ± 0.54 %ID/g (*p* < 0.0001)) (Appendix A). With respect to the other healthy organs and tissues, [^68^Ga]Ga-BL31 exhibited a similar pattern of uptake as compared to [^68^Ga]Ga-BL02. While it had a higher kidney uptake than [^68^Ga]Ga-BL25 (2.23 ± 0.21 vs. 1.92 ± 0.07 %ID/g, *p* = 0.0091) at 2 h p.i., its higher tumor uptake provided a higher tumor-to-kidney ratio than [^68^Ga]Ga-BL25 (3.95 ± 0.54 vs. 2.92 ± 0.80 (*p* = 0.0022)). Overall, [^68^Ga]Ga-BL31 had a lower kidney uptake, while retaining an overall desirable pharmacokinetic profile that was similar to [^68^Ga]Ga-BL02. This was further confirmed by PET/CT images of [^68^Ga]Ga-BL31 in Daudi-bearing xenograft mice, wherein the kidneys were much less visible in comparison to the PET image of [^68^Ga]Ga-BL02. A comparison of LogD_7.4_ between [^68^Ga]Ga-BL31 (−4.17 ± 0.14) and [^68^Ga]Ga-BL02 (−4.20 ± 0.44) showed no statistically significant difference.

### 3.4. Comparison of [^68^Ga]Ga-BL31 with [^68^Ga]Ga-Pentixafor

Previous comparison between [^68^Ga]Ga-BL02 and [^68^Ga]Ga-Pentixafor showed that [^68^Ga]Ga-BL02 had an overall superior imaging contrast as compared to [^68^Ga]Ga-Pentixafor, based on their biodistribution profile and assessment of PET images of Daudi xenograft-bearing mice. Given that [^68^Ga]Ga-BL31 has a similar biodistribution profile as that of [^68^Ga]Ga-BL02, a similar improvement in imaging contrast was found in the comparison between [^68^Ga]Ga-BL31 and [^68^Ga]Ga-Pentixafor. Inspection of the PET images of [^68^Ga]Ga-BL31 and [^68^Ga]Ga-Pentixafor at 1 and 2 h p.i. showed a higher tumor uptake and a relatively lower uptake by surrounding tissues (Figure 5). This was validated by the ex vivo biodistribution studies. [^68^Ga]Ga-BL31 showed a higher tumor uptake than [^68^Ga]Ga-Pentixafor at 1 h (9.41 ± 1.04 vs. 6.31 ± 1.02 (*p* < 0.001)) and 2 h (8.97 ± 1.47 vs. 5.24 ± 0.43 (*p* < 0.001)) p.i. With respect to non-target organs, [^68^Ga]Ga-BL31 showed a lower uptake than [^68^Ga]Ga-Pentixafor in organs such as the blood pool (1 h: 0.36 ± 0.07 vs. 0.97 ± 0.12 (*p* < 0.0001) and 2 h: 0.08 ± 0.04 vs. 0.28 ± 0.04 (*p* < 0.0001)); muscle (1 h: 0.08 ± 0.02 vs. 0.20 ± 0.04 (*p* < 0.0001) and 2 h: 0.03 ± 0.01 vs. 0.06 ± 0.02 (*p* = 0.016)); liver (1 h: 0.67 ± 0.10 vs. 1.23 ± 0.15 (*p* < 0.0001) and 2 h: 0.66 ± 0.10 vs. 0.89 ± 0.12 (*p* = 0.0020)); and lungs (1 h: 0.56 ± 0.09 vs. 1.16 ± 0.14 (*p* < 0.0001) and 2 h: 0.26 ± 0.05 vs. 0.54 ± 0.08 (*p* < 0.0001)). This translated into higher tumor-to-organ ratios and imaging contrast. [^68^Ga]Ga-BL31 showed an equivocal or higher kidney uptake than [^68^Ga]Ga-Pentixafor (2 h: 2.28 ± 0.21 vs. 1.76 ± 0.38 (*p* = 0.012)). Given the higher tumor uptake of [^68^Ga]Ga-BL31, however, [^68^Ga]Ga-BL31 had a higher tumor-to-kidney ratio than [^68^Ga]Ga-Pentixafor (1 h: 3.73 ± 0.52 vs. 2.29 ± 0.38 (*p* = 0.0002) and 2 h: 3.95 ± 0.54 vs. 3.07 ± 0.54 (*p* = 0.01)).

## 4. Discussion

Herein, we report an optimized radiopharmaceutical, [^68^Ga]Ga-BL31, with improved tumor-to-kidney contrast ratios, with superior imaging capabilities as compared to the leading clinical CXCR4-targeting radiotracer, [^68^Ga]Ga-Pentixafor. Reduction in non-target kidney uptake has been a significant focus in the development of radiopharmaceuticals of peptides. Kidney uptake in several leading clinical radiopharmaceuticals, such as [^68^Ga]Ga-/[^177^Lu]Lu-DOTA-TATE, results in decreased tumor-to-kidney contrast and dose limitations, as deposition of ionizing radioactivity can lead to acute kidney injury and potentially chronic kidney disease or renal failure [34]. For example, the retention of SSTR_2_-targeting radioligand therapeutics in the kidney via megalin-mediated proximal tubular reabsorption have necessitated strategies such as dosimetry studies to avoid off-target nephrotoxicity, while maximizing the administered dose [34]. Other peptide-based radiopharmaceuticals have also shown to be resorbed in the proximal tubules, resulting in the high retention of ionizing radiation in the kidneys [35,36,37,38]. Specific to CXCR4-targeting radiopharmaceuticals, *VHL*-mutant malignancies such as ccRCC have high expressions of CXCR4. ccRCC generally has a poor prognosis with limited treatment options and no biomarkers for risk stratifications; as such, non-invasive molecular imaging and radioligand therapy targeting CXCR4 may prove to be a viable strategy with radiotracers with a sufficiently low uptake in the kidney parenchyma [39]. Therefore, our study directly addresses the ongoing issue of the off-target kidney retention of peptide-based radiopharmaceuticals.

The kidney uptake of radiopharmaceuticals is likely mediated by multiple mechanisms, depending on the nature of the targeting vector. For example, the renal brush border of the kidney has been shown to be negatively charged; therefore, the overall charge of the radiopharmaceutical may play a key role in determining the rate of reabsorption into the renal proximal tubular cells [36,40]. To mitigate kidney uptake, strategies such as the administration of positively charged amino acids, gelatin-based plasma expanders such as gelofusine, and the use of linkers that are cleaved by kidney-specific proteases, have been explored as possible strategies [35,41,42]. However, the logistics that are involved in the administration of additional agents, and issues with both species and patient variability in the activity and specificity of kidney proteases, reduce the appeal of these approaches. As such, strategic modification of the chemical structure of the radiopharmaceutical to improve its pharmacokinetic parameters while reducing off-target binding in the kidneys is the ideal strategy.

We targeted the linker and chelator as modifiable sites for decreased kidney uptake and the optimization of other pharmacokinetic parameters, given that the pharmacophore has been extensively optimized. Linker modifications play a crucial role in modulating the kidney accumulation of radiotracers. For example, the addition of an anionic polyglutamate- and glutamic acid-based linker to a minigastrin analogue and RGD-based radiotracer, respectively, effectively reduced kidney uptake [40,43]. The charge on the SSTR_2_-targeting ligands likely play a key role in their retention in the kidney [26,44,45]. The optimization of tumor-to-kidney ratios would considerably enhance the viability and applicability of our CXCR4-targeting radiopharmaceutical, considering that both [^68^Ga]Ga-BL02 and [^18^F]BL08 showed comparable, if not superior, PET imaging capabilities as compared to other CXCR4-targeting radiotracers [24,28,46].

For consistency between previous studies, we used the Daudi Burkitt lymphoma xenograft model, which has shown to have robust CXCR4 and comparable CXCR4 expression as compared to other hematological malignancies [24]. We selected a piperidine-based (Pip) linker as our cationic hydrophilic linker, as previous studies showed that the use of the Pip linker in a derivative of the α-melanocyte-stimulating hormone to target the melanocortin-1 receptor resulted in not only improved tumor uptake, but also higher tumor-to-organ ratios [29]. However, [^68^Ga]Ga-BL06 was clearly inferior to [^68^Ga]Ga-BL02, with worse clearance rates, higher uptake in non-target organs, and worse uptake in Daudi xenografts. While our original hypothesis was that the hydrophilic nature of the Pip linker would result in high-contrast PET images from relatively rapid excretion rates, our results indicate the opposite. In theory, the ammonium salt of *N*-methylpiperidine has a pK_a_ of approximately 10.08, implying protonation of the piperidine-based linker at physiological pH, which should increase clearance rates. An explanation may be that the additional cationic charge results in a stronger interaction with erythrocytes, reducing the rate of extravasation of the radiotracer from the bloodstream into the tumor tissue; a similar effect is seen with [^68^Ga]Ga-/[^177^Lu]Lu-BL01, which is highly positively charged [47]. While a potential explanation for spleen and liver uptake is from mCXCR4-mediated interactions, the LY2510924 pharmacophore has a >1000-fold decrease in affinity to mCXCR4 as compared to hCXCR4, ruling out this explanation [48]. These results further reflect the subtle but important role that linker pharmacokinetic effects play in determining tumor uptake.

The modification of chelators has shown previously to alter both the binding and pharmacokinetic properties of radiotracers, thereby changing their imaging properties. Indeed, the use of a trifluoroborate-based radiolabeling moiety ([^18^F]BL08) over the radiometal-chelated DOTA ([^68^Ga]Ga-BL02) led to comparatively higher tumor-to-kidney ratios as compared to [^68^Ga]Ga-Pentixafor [28]. DOTAGA, possessing an additional carboxylate arm, is primarily used to enhance the hydrophilicity of radiotracers [31,49]. However, [^68^Ga]Ga-BL30 showed reduced uptake in the Daudi xenografts, potentially from unfavorable interactions between the negatively charged metal–chelator complex with the receptor. Furthermore, there was a concurrent increase in kidney uptake [27]. As such, further modifications of the metal-chelating radioprosthetic group were not undertaken.

On the other hand, our strategy to fine-tune the anionic triglutamate linker with similar negatively charged amino acids to reduce kidney uptake was more successful. Both aspartate and homoglutamate are close structural analogues to glutamate, bearing one more or less methylene in its side chain, respectively. Previous work on ^67^Ga-based polyaspartate and polyglutamate bone imaging agents showed a minimal kidney uptake in the polyaspartate variant, while the polyglutamate-based radiotracer had a comparatively higher kidney uptake [27]. While [^68^Ga]Ga-BL17 did not show any reduction in kidney uptake, [^68^Ga]Ga-BL25 showed a significantly lower kidney uptake at 2 h p.i., which validated our hypothesis that incorporating structural analogues in our linkers may alter kidney uptake. However, [^68^Ga]Ga-BL25 had a lower uptake in the Daudi xenografts, resulting in poorer tumor-to-organ ratios and lowered imaging contrast.

The use of D-amino acids is a well-validated strategy to enhance the metabolic stability of many peptides and peptidomimetic molecules, as many proteins do not recognize the chiral counterparts of the recognition sequence [50]. We theorized that incorporating the D-isomer of glutamate in our linker may potentially ablate any receptor-mediated kidney uptake, while still maintaining the same physiochemical properties as the original triglutamate linker. Ex vivo biodistribution studies of [^68^Ga]Ga-BL20 confirmed that it possesses the same overall pharmacokinetic properties as [^68^Ga]Ga-BL02, as expected of the opposing chiral linker with similar stability. Unfortunately, there was also no reduction in kidney uptake. These results indicate that the specific kidney uptake in our CXCR4-targeting radiotracer is likely not specific for the L-isomers of anionic amino acids.

Given the lowered kidney uptake seen in the aspartate linker, we explored the use of cysteic acid, a bioisostere of aspartate with a sulfonate in place of the carboxylate, in order to mimic the drop in kidney uptake while retaining the high uptake in the Daudi xenografts, as seen in [^68^Ga]Ga-BL02. Gratifyingly, [^68^Ga]Ga-BL31 exhibited a lower kidney uptake while retaining the high uptake in the Daudi xenograft, comparable to [^68^Ga]Ga-BL02. No other significant absolute differences in the biodistribution were found between the two radiotracers. While [^68^Ga]Ga-BL31 still had a higher kidney uptake value than [^68^Ga]Ga-Pentixafor, the relatively small difference, coupled with [^68^Ga]Ga-BL31’s significantly higher tumor uptake, resulted in higher tumor-to-kidney ratios for [^68^Ga]Ga-BL31. As such, the higher ratio of [^68^Ga]Ga-BL31 enables the higher imaging contrast of lesions that are located proximally to the kidneys on PET imaging, even though the absolute kidney uptake is greater. Combined with its lower uptake in non-target organs as compared to [^68^Ga]Ga-Pentixafor, [^68^Ga]Ga-BL31 represents an improved candidate for targeted CXCR4 PET imaging as compared to both [^68^Ga]Ga-BL02 and [^68^Ga]Ga-Pentixafor, with potential for clinical translation. With respect to therapeutic applications, there is a concern that the conjugation of another radiometal may abrogate the affinity of the radiotheranostic to the target receptor. This is shown with [^68^Ga]Ga-Pentixafor and [^177^Lu]Lu-Pentixather. We have previously demonstrated that [^177^Lu]Lu-BL02 shows no loss of performance with respect to both tumor uptake and non-target organ clearance; however, an assessment of both in vitro binding affinity and in vivo performance is needed to definitively verify the applicability of [^177^Lu]Lu-BL31.

There are a few potential explanations regarding the reduction in kidney uptake. The placement of three consecutive charged amino acids may increase the pK_a_ of the neighboring carboxylate due to unfavorable electrostatic repulsions. The sulfonate has a lower pK_a_ than a carboxylate (approximately −1.7 versus 1.9 in aspartate), and as such, may be able to better retain its negative charge, thereby increasing the excretion of the radiotracer from the kidneys. However, there was no difference in LogD_7.4_ values between [^68^Ga]Ga-BL02 and [^68^Ga]Ga-BL31. The increased capacity to hold a negative charge may also increase the repulsion between the negatively charged brush border, reducing the reabsorption of [^68^Ga]Ga-BL31. Alternatively, the loss of the methylene group as compared to the glutamate may reduce the interactions between the linker and receptors in the kidney, lowering the specific uptake of the radiotracer to the kidneys. This phenomenon was also seen with the Asp-based linker. Finally, as opposed to carboxylic acids, sulfonic acids are non-planar in structure and are more polar and acidic, which can generate stronger and additional polar contacts with the desired target [51]. However, this non-planar structure may lead to a potential loss of contacts, which may alter the molecular recognition implicated in the kidney-specific receptor-mediated uptake of the radiotracer [52]. For example, cysteic and homocysteic acids are known to act as neurotransmitters in the central nervous system and have differential binding and downstream properties as compared to their carboxylate counterparts [53,54]. However, more work must be undertaken to prove the underlying mechanism of reduced kidney retention. Nonetheless, these results highlight the potential of sulfonates and other carboxylate bioisosteres to modulate important pharmacokinetic parameters of radiopharmaceuticals for maximal imaging contrast and radioligand therapeutic index.

## 5. Conclusions

In summary, [^68^Ga]Ga-BL31, containing a cysteic acid-based linker, is shown to have improved tumor-to-kidney ratios as compared to that of [^68^Ga]Ga-BL02, while retaining a favorable biodistribution profile for high contrast PET imaging. In a direct comparison, [^68^Ga]Ga-BL31 had a higher tumor uptake and tumor-to-organ contrast ratios as compared to that of [^68^Ga]Pentixafor. As such, [^68^Ga]Ga-BL31 is a promising candidate for radiotheranostic applications in a clinical setting.

## 6. Patents

D. Kwon, Z. Zhang, K.-S. Lin and F. Benard submitted a patent application (PCT/CA2020/050521) based on some of the outlined work. This patent has been licensed by the University of British Columbia and BC Cancer to Alpha-9 Theranostics.

## Figures and Tables

**Figure 1 pharmaceutics-14-01502-f001:**
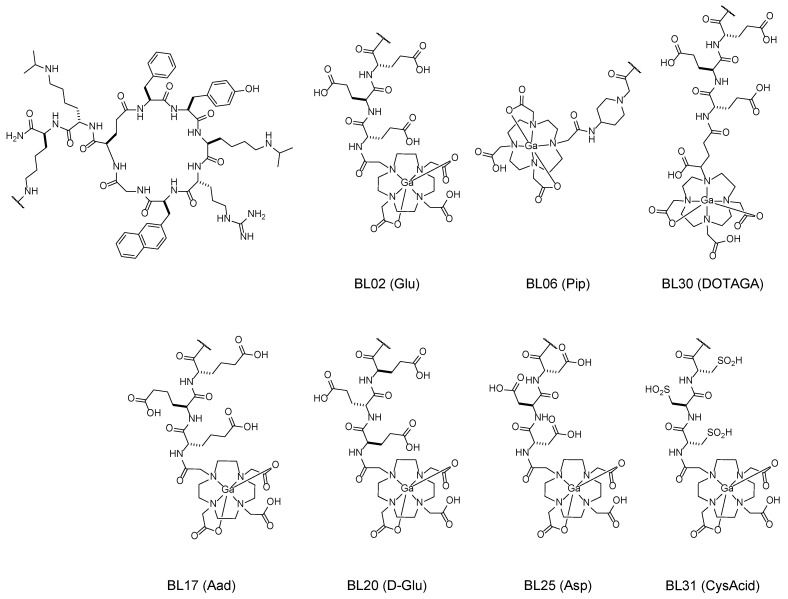
Representative chemical structures of the gallium-labeled BL02 (Glu); BL06 (Pip); BL17 (Aad); BL20 (D-Glu); BL25 (Asp); BL30 (DOTAGA); and BL31 (CysAcid).

**Figure 2 pharmaceutics-14-01502-f002:**
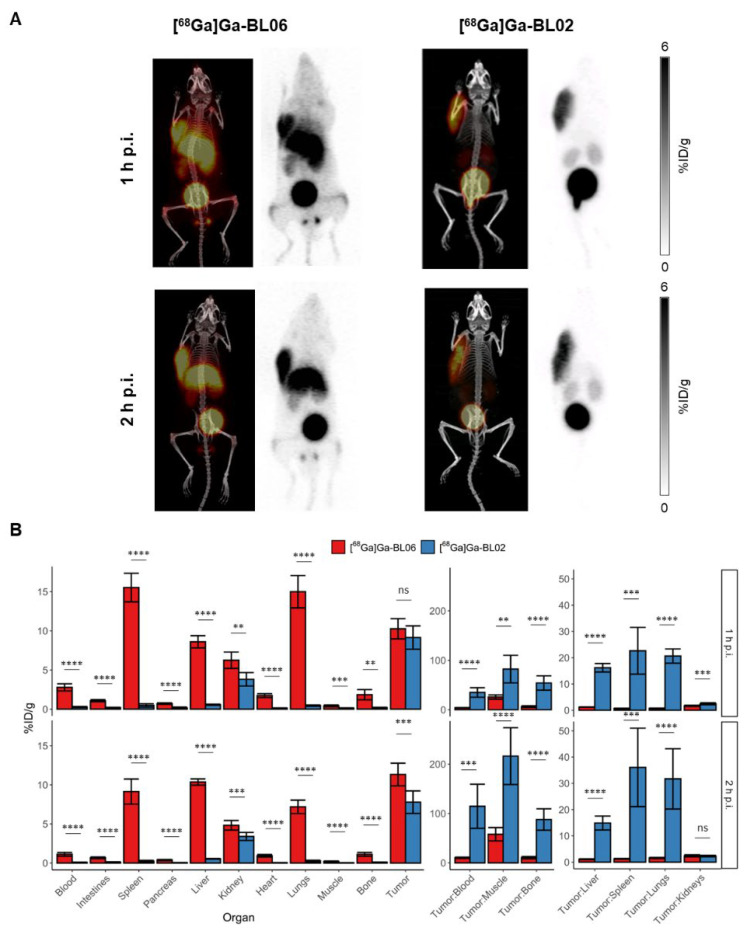
(**A**) Maximum intensity projections for PET/CT and PET alone at 1 and 2 h p.i. of [^68^Ga]Ga-BL06 and [^68^Ga]Ga-BL02. Scales of the PET images of [^68^Ga]Ga-BL02 and [^68^Ga]Ga-BL06 are 0–6 %ID/g. (**B**) Ex vivo biodistribution data of [^68^Ga]Ga-BL06 in comparison to that of [^68^Ga]Ga-BL02 in Daudi xenograft-bearing mice at 1 and 2 h p.i. (*p* < 0.01 = **, *p* < 0.001 = ***, *p* < 0.0001 = ****, ns = not significant).

**Figure 3 pharmaceutics-14-01502-f003:**
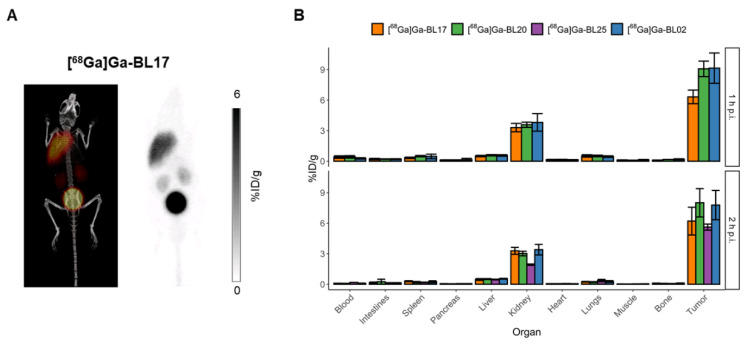
(**A**) Maximum intensity projections for PET/CT and PET alone at 1 h p.i. [^68^Ga]Ga-BL17. Scales of the PET images of [^68^Ga]Ga-BL17 are 0–6 %ID/g. (**B**) Ex vivo biodistribution data of [^68^Ga]Ga-BL17; [^68^Ga]Ga-BL20; and [^68^Ga]Ga-BL25 in comparison to that of [^68^Ga]Ga-BL02 in Daudi xenograft-bearing mice at 1 and 2 h p.i. [^68^Ga]Ga-BL25 was only evaluated at 2 h p.i.

**Figure 4 pharmaceutics-14-01502-f004:**
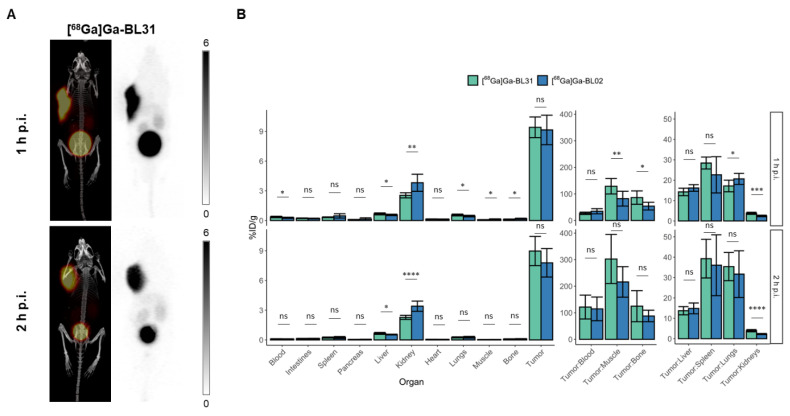
(**A**) Maximum intensity projections for PET/CT and PET alone at 1 and 2 h p.i. of [^68^Ga]Ga-BL31. Scales of the PET images of [^68^Ga]Ga-BL31 are 0–6 %ID/g. (**B**) Ex vivo biodistribution data of [^68^Ga]Ga-BL31 in comparison to that of [^68^Ga]Ga-BL02 in Daudi xenograft-bearing mice at 1 and 2 h p.i. (*p* < 0.05 = *, *p* < 0.01 = **, *p* < 0.001 = ***, *p* < 0.0001 = ****, ns = not significant).

**Figure 5 pharmaceutics-14-01502-f005:**
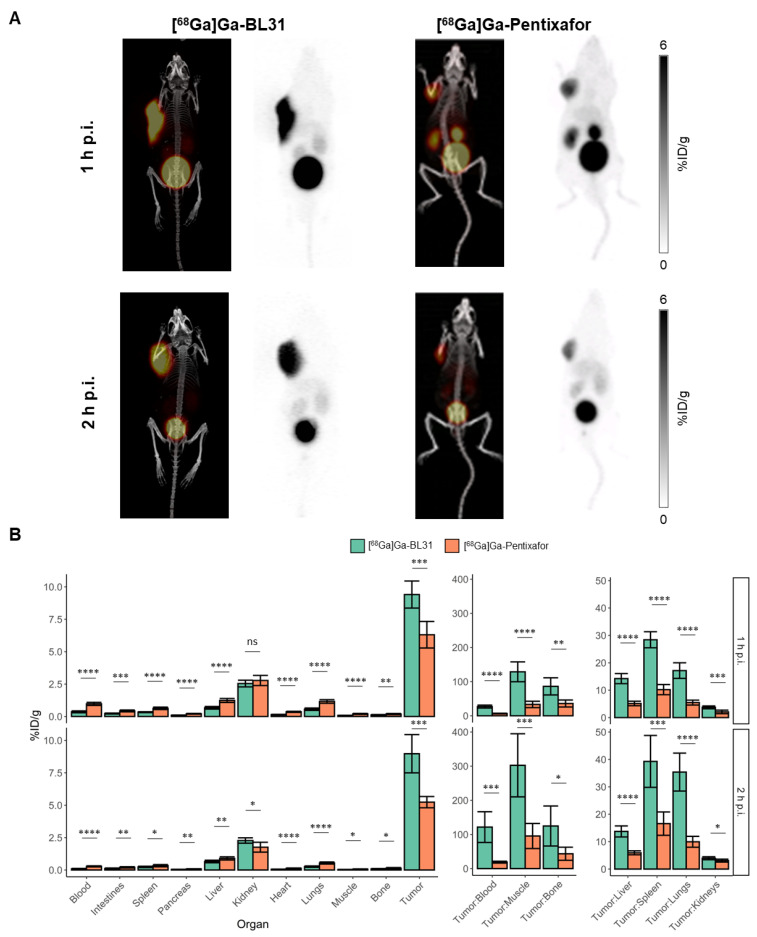
(**A**) Maximum intensity projections for PET/CT and PET alone at 1 and 2 h p.i. of [^68^Ga]Ga-BL31 and [^68^Ga]Ga-Pentixafor. Scales of the PET images of [^68^Ga]Ga-BL31 and [^68^Ga]Ga-Pentixafor are 0–6 %ID/g. (**B**) Ex vivo biodistribution data of [^68^Ga]Ga-BL31 in comparison to that of [^68^Ga]Ga-Pentixafor in Daudi xenograft-bearing mice at 1 and 2 h p.i. (*p* < 0.05 = *, *p* < 0.01 = **, *p* < 0.001 = ***, *p* < 0.0001 = ****, ns = not significant).

**Table 1 pharmaceutics-14-01502-t001:** In vitro data of each ^68^Ga-labeled radiotracer. Peptide net charge was calculated at pH 7.40. All values are in triplicate, unless indicated otherwise.

Radiotracer	IC_50_ (nM)	Peptide Net Charge	LogD_7.4_
[^68^Ga]BL02	27.9 ± 12.5 ^†^	3.0	−4.20 ± 0.44
[^68^Ga]BL06	26.3 ± 27.6 ^†^	4.0	N.D.
[^68^Ga]BL17	13.0 ± 8.6 ^†^	3.0	N.D.
[^68^Ga]BL20	N.D.	3.0	N.D.
[^68^Ga]BL25	21.3 ± 0.1	3.0	N.D.
[^68^Ga]BL30	22.7 ± 1.2	2.0	N.D.
[^68^Ga]BL31	16.2 ± 4.2	3.0	−4.17 ± 0.14

^†^*n* = 4, N.D.: not determined.

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
