# Peer review of "Reducing the Kidney Uptake of High Contrast CXCR4 PET Imaging Agents via Linker Modifications"

_pharmaceutics, 2022, doi:10.3390/pharmaceutics14071502_

Round 1

Reviewer 1 Report

The manuscript by Kwon and coauthors describes the synthesis and preclinical evaluation of a series of LY2510924-based, CXCR4 targeted radiopharmaceuticals with different linkers and two alternative chelators. Out of this series of compounds, one analog with improved in vivo characteristics, namely [68Ga]Ga-BL31, shows promise for further evaluation within the context of a CXCR4-targeted theranostic approach.

The presented experimental data are comprehensive and support the authors’ conclusions. However, in general, the theoretical background, working hypothesis and discussion of the data lack depth and precision. This also applies to the quality of the scientific writing/English language. Thus, a major revision of the manuscript is warranted.

Specific points:

Line 31: “is the” must be “are”

Line 36: Please rephrase “more so than” to correct English

Line 36/37: Please rephrase – as such, the sentence is not clear. Should be something like “ … the in vivo performance of … largely depends on optimized ligand pharmacokinetics, adapted to the respective application (imaging vs therapy).”

Line 42/43: Please rephrase – this sentence is scientifically unprecise.

Line 51: metastasis as a process cannot be “upregulated” – enhanced may be more appropriate.

Line 56: please replace “disclosed” by “developed” or another expression.

Lines 62-73: This paragraph needs more depth and precision. It is well known, that the tubular reuptake of peptide radiotracers by the megalin/cubilin complex is driven by positive net charge, which may already explain the statement in lines 66/67. It is thus not understandable why the authors opt for a cationic linker in the first place – the piperidine linker may only serve as a further confirmation of enhanced tracer reabsorption with increasing peptide net charge. Please thoroughly revise the rationale for designing the tracers evaluated in this study.

Line 145: Please add the molar amount of injected peptide in brackets after “4-7 MBq”

Line 148: “anaesthetized” instead of “sedated”

Line 158: See comment Line 145

Lines 184-192: Please omit. If not related to the lead compound of the study, BL31, synthetic details are of minor interest to the reader.

Table 1: Why was the affinity of BL20 not determined? This value is of interest, especially since the authors claim “ablation of receptor mediated kidney uptake” (line 379) by this modification. Please add the missing data. Furthermore, please move the information on the molar activity, the RCP and the RCY to the supplemental section, but rather add relevant physicochemical data such as the log D and the peptide net charge at pH 7.4 to the table. These are parameters of major interest for kidney uptake and thus highly relevant for the evaluation of the ligands in this study.

Figures 2,3,4 and 5: Please scale all PET images to the same maximum value (%iD/g) – this allows a much more direct and precise visual distinction of the different tumor- and kidney uptakes of the tracers investigated.

Discussion, line 309: please replace “disclose” by another expression

Lines 331-315: Please specifically refer to therapeutic radiopharmaceuticals and explain in more detail, why high kidney uptake is a problem.

Line 314: PSMA-ligands are not a good example in this context, because their uptake in the kidney relies primarily on endogenous PSMA-expression in the kidney, and thus “on-target-off-tissue” binding of the radiopharmaceuticals. A much better example for non-specific kidney uptake of targeted therapeutic radioligands (which is what this study also aims to address, given that there is virtually no CXCR4 expression in the human kidney) are therapeutically used sst2-ligands such as DOTATATE. Please thoroughly revise this paragraph in clearly stating what the aim of this study was (reducing NONSPECIFIC kidney uptake of the BL-ligands) and cite relevant literature to underline your statements.

Line 329: “make both strategies less viable” : this is not true; both strategies are successful in the clinic, where other options (modifications of the targeting vector) are not available or hard to implement. Please rephrase.

Line 337: please also cite the work on sst-ligands, where it has been unambiguously shown that peptide NET CHARGE is a major determinant of kidney accumulation for radiolabeled sst-ligands.

Line 338: please replace “viability” by a correct expression.

Line 345: please rephrase “clear non-superiority” – this is an unusual expression.

Line 354: the word “radioprosthesis” does not exist. Please replace by “radiolabel”, and please replace “radiometal-chelated DOTA radioprosthesis” by “DOTA-radiometal chelate”.

Line 358: if the decrease of the affinity of BL-30 compared to the other compounds was not statistically significant, why is it mentioned here?

Line 362: “to elements in the kidney tubules” – which elements? This is pure speculation, please omit.

Lines 365-366: please rephrase sentence to more scientific language (“our” anionic linker, “was met with a greater degree of success”).

Lines 379: “may potentially ablate any receptor-mediated kidney uptake”? Which “receptor”? Why would a change in stereochemistry in a linker, that obviously does not interact with the CXCR4 binding pocket (given the same performance of BL-02 and BL-20 in vivo), change kidney uptake ???

Lines 407/408: which interactions between the linker and “receptors in the kidney”?

Lines 410-419: please omit (or carefully rephrase) this paragraph – it is speculative and strangely out of context here. The only valid (and sufficient) explanation for the superiority of the sulfonic acid linker is its lower pKa.

General comment: a discussion on the potential reasons for the very different in vivo behavior of BL-06 compared to BL-02 is completely missing. Apparently, uptake of BL-06 in all mouse-CXCR4 expressing organs is substantially enhanced (lung, liver, spleen, bone), which hints towards enhanced mCXCR4 affinity of the compound with the cationic linker, especially since its log D seems to be unchanged and thus cannot be the reason for a change in organ distribution. The authors should provide binding data to mCXCR4 or at least critically discuss this finding.

Author Response

The manuscript by Kwon and coauthors describes the synthesis and preclinical evaluation of a series of LY2510924-based, CXCR4 targeted radiopharmaceuticals with different linkers and two alternative chelators. Out of this series of compounds, one analog with improved in vivo characteristics, namely [68Ga]Ga-BL31, shows promise for further evaluation within the context of a CXCR4-targeted theranostic approach.

The presented experimental data are comprehensive and support the authors’ conclusions. However, in general, the theoretical background, working hypothesis and discussion of the data lack depth and precision. This also applies to the quality of the scientific writing/English language. Thus, a major revision of the manuscript is warranted.

We thank the authors for their constructive comments regarding our manuscript, with respect to both the scientific background and aims, and the quality of the writing. The grammatical and writing points listed below have been addressed and summarily corrected in the revised version of the manuscript:

Specific points:

Line 31: “is the” must be “are”

Because the verb “is” refers to a singular noun “consideration”, the verb conjugation should be singular (is) and not plural (are).

Line 36: Please rephrase “more so than” to correct English

Line 36/37: Please rephrase – as such, the sentence is not clear. Should be something like “ … the in vivo performance of … largely depends on optimized ligand pharmacokinetics, adapted to the respective application (imaging vs therapy).”

We have modified the sentence to as follows: “Radiopharmaceuticals for imaging and therapy are highly reliant on their pharmacokinetic properties for maximal imaging contrast and therapeutic index” which highlights the importance of pharmacokinetics to maximize contrast and therapeutic index, as explained in the rest of the paragraph. 

Line 42/43: Please rephrase – this sentence is scientifically unprecise.

We have modified the sentence to the following: “The difference in measured radioactivity between the diseased tissue and surrounding area is vital for maximal imaging contrast. Similarly, enhancing the uptake of a radiotracer in target lesions while reducing uptake in healthy tissue will reduce off-target radiotoxicity while maximizing therapeutic outcomes.” We believe this clarifies the importance of imaging contrast and therapeutic index in the development of radiopharmaceuticals.

Line 51: metastasis as a process cannot be “upregulated” – enhanced may be more appropriate.

We have changed the wording to “enhancing the metastatic potential of cancer cells.”

Line 56: please replace “disclosed” by “developed” or another expression.

We have replaced the word “disclosed” with “reported.”

Lines 62-73: This paragraph needs more depth and precision. It is well known, that the tubular reuptake of peptide radiotracers by the megalin/cubilin complex is driven by positive net charge, which may already explain the statement in lines 66/67. It is thus not understandable why the authors opt for a cationic linker in the first place – the piperidine linker may only serve as a further confirmation of enhanced tracer reabsorption with increasing peptide net charge. Please thoroughly revise the rationale for designing the tracers evaluated in this study.

Our motivation for using a cationic linker was because it was not a priori knowledge that a cationic linker would increase kidney uptake. We were guided by our previous work in developing radiopharmaceuticals targeting the MC1 receptor (Zhang et al. Theranostics 2017) which showed that the addition of a piperidine-based cationic linker increased the tumor-to-kidney ratios. While not containing as many cationic amino acids as the LY2510924 peptide, the alpha-MSH-based pharmacophore had an overall positive charge with the presence of an Arg residue.

As such, we have added this reference and clarified our rationale in adding a piperidine-based linker in our radiopharmaceutical as shown below:

“This was motivated by our previous work in the development of MC1R-targeting radiopharmaceuticals, which showed that a piperidine-based linker was essential in increasing the tumor-to-kidney ratios.”

Line 145: Please add the molar amount of injected peptide in brackets after “4-7 MBq”

The molar amount “(10-700 pmol)” was inserted in the text.

Line 148: “anaesthetized” instead of “sedated”

Line 158: See comment Line 145

The wording has been changed as suggested.

Lines 184-192: Please omit. If not related to the lead compound of the study, BL31, synthetic details are of minor interest to the reader.

As suggested, the synthetic details have been removed from the body of the text. Specifics on the synthesis of the radiopharmaceuticals are available in the supplementary information.

Table 1: Why was the affinity of BL20 not determined? This value is of interest, especially since the authors claim “ablation of receptor mediated kidney uptake” (line 379) by this modification. Please add the missing data. Furthermore, please move the information on the molar activity, the RCP and the RCY to the supplemental section, but rather add relevant physicochemical data such as the log D and the peptide net charge at pH 7.4 to the table. These are parameters of major interest for kidney uptake and thus highly relevant for the evaluation of the ligands in this study.

Because we demonstrated that the change in the tri-anionic amino acid-based linkers did not show a significant change in binding affinity to CXCR4, we opted not to measure the binding affinity of the D-glutamic acid-based radiopharmaceutical given that [68Ga]Ga-BL20 did not show a significant change from [68Ga]Ga-BL02. We do not claim that [68Ga]Ga-BL20 ablates kidney uptake but rather, mention that: “For the kidneys, [68Ga]Ga-BL20 showed no statistically significant difference in uptake compared to [68Ga]Ga-BL02 (3.58±0.26 and 3.02±0.22 %ID/g at 1 and 2 h p.i., respectively).”

As suggested, the molar activity, the RCP and RCY moved to the supplemental. The measured partition coefficients and peptide net charge at pH 7.40 was calculated and put in the table.

Figures 2,3,4 and 5: Please scale all PET images to the same maximum value (%iD/g) – this allows a much more direct and precise visual distinction of the different tumor- and kidney uptakes of the tracers investigated.

As suggested, the PET images were scaled to the same maximal value from 0 – 6.

Discussion, line 309: please replace “disclose” by another expression

The word “disclose” was replaced with “report.”

Lines 331-315: Please specifically refer to therapeutic radiopharmaceuticals and explain in more detail, why high kidney uptake is a problem.

Line 314: PSMA-ligands are not a good example in this context, because their uptake in the kidney relies primarily on endogenous PSMA-expression in the kidney, and thus “on-target-off-tissue” binding of the radiopharmaceuticals. A much better example for non-specific kidney uptake of targeted therapeutic radioligands (which is what this study also aims to address, given that there is virtually no CXCR4 expression in the human kidney) are therapeutically used sst2-ligands such as DOTATATE. Please thoroughly revise this paragraph in clearly stating what the aim of this study was (reducing NONSPECIFIC kidney uptake of the BL-ligands) and cite relevant literature to underline your statements.

As per the reviewers’ recommendations, we have elaborated on the impact of nephrotoxicity on leading clinical radiopharmaceuticals used for radioligand therapy:

Kidney uptake in several leading clinical radiopharmaceuticals, such as [68Ga]Ga-/[177Lu]Lu-DOTA-TATE, results in decreased tumor-to-kidney contrast  and dose limitations, as deposition of ionizing radioactivity can lead to acute kidney injury and potentially chronic kidney disease or renal failure.1 34For example, the retention of SSTR2-targeting radioligand therapeutics in the kidney via megalin-mediated proximal tubular reabsorption have necessitated strategies such as dosimetry studies to avoid off-target nephrotoxicity while maximizing the administered dose.1 Other peptide-based radiopharmaceuticals have shown to be resorbed in the proximal tubules also, resulting in high retention of ionizing radiation in the kidneys.2–5 Specific to CXCR4-targeting radiopharmaceuticals, VHL-mutant malignancies such as ccRCC have high expressions of CXCR4. ccRCC generally has a poor prognosis with limited treatment options, and no biomarkers for risk stratifications; non-invasive molecular imaging and radioligand therapy targeting CXCR4 may prove to be a viable strategy with radiotracers with sufficiently low uptake in the kidney parenchyma.6  Therefore, our study directly addresses an ongoing issue of off-target kidney retention of peptide-based radiopharmaceutical.

Line 329: “make both strategies less viable” : this is not true; both strategies are successful in the clinic, where other options (modifications of the targeting vector) are not available or hard to implement. Please rephrase.

The phrase “make both strategies less viable” has been changed to “reduce the appeal of these approaches.”

Line 337: please also cite the work on sst-ligands, where it has been unambiguously shown that peptide NET CHARGE is a major determinant of kidney accumulation for radiolabeled sst-ligands.

This has been noted with the following sentence: “The charge on the SSTR2-targeting ligands likely play a key role in their retention in the kidney.7–9

Line 338: please replace “viability” by a correct expression.

The word “viability” has been replaced with “efficacy.”

Line 345: please rephrase “clear non-superiority” – this is an unusual expression.

The phrase has been replaced with “was clearly inferior to that of”

Line 354: the word “radioprosthesis” does not exist. Please replace by “radiolabel”, and please replace “radiometal-chelated DOTA radioprosthesis” by “DOTA-radiometal chelate”.

Respectfully, a search of the literature reveals the use of “radioprosthesis” “radioprosthetic” and “prosthetic” as appropriate descriptors of various radiolabeled chemical conjugates.10–14 Because the generality of this statement extends beyond radiometal-chelate conjugates (e.g. 18F- or 11C-based radioconjugates), we believe this descriptor is appropriate in the context of the development of radiopharmaceuticals beyond radiometal applications.

Line 358: if the decrease of the affinity of BL-30 compared to the other compounds was not statistically significant, why is it mentioned here?

The aforementioned information has been removed from the discussion.

Line 362: “to elements in the kidney tubules” – which elements? This is pure speculation, please omit.

The aforementioned information has been removed from the discussion.

Lines 365-366: please rephrase sentence to more scientific language (“our” anionic linker, “was met with a greater degree of success”).

The “our” has been changed to “the” and “was met with a greater degree of success” was changed to “was more successful.”

Lines 379: “may potentially ablate any receptor-mediated kidney uptake”? Which “receptor”? Why would a change in stereochemistry in a linker, that obviously does not interact with the CXCR4 binding pocket (given the same performance of BL-02 and BL-20 in vivo), change kidney uptake ???

Lines 407/408: which interactions between the linker and “receptors in the kidney”?

As noted in the introduction and past text in the discussion, the charge of radiopharmaceuticals and kidney-based receptors such as the megalin receptor are relevant in the retention of peptide-based radiopharmaceuticals in the kidney, as the reviewers have reminded us earlier. With the possibility that the linker, a highly hydrophilic and charged element, may interact with receptors implicated in peptide retention, we believed that the use of D-Glu and the cysteic acid isostere over L-Glu may reduce those interactions. As our citations note, both D-amino acid-based and sulfonate-based strategies have been shown to successfully abrogate putative interactions.

Lines 410-419: please omit (or carefully rephrase) this paragraph – it is speculative and strangely out of context here. The only valid (and sufficient) explanation for the superiority of the sulfonic acid linker is its lower pKa.

We agree with the reviewers that the lower pKa is the more likely explanation; however, due to the lack of difference in the LogD7.4 between Ga-BL02 and Ga-BL31, it is expected that the discussion include some other possibilities as to this difference. As noted in our discussion and our citations, the sulfonate, based on first principles, is structurally and electrochemically different than that of the carboxylate and therefore, can mediate unique receptor-mediated interactions, though possessing a similar charge. We therefore added the sentence “However, more work must be undertaken to definitively prove the underlying mechanism of reduced kidney retention.”

General comment: a discussion on the potential reasons for the very different in vivo behavior of BL-06 compared to BL-02 is completely missing. Apparently, uptake of BL-06 in all mouse-CXCR4 expressing organs is substantially enhanced (lung, liver, spleen, bone), which hints towards enhanced mCXCR4 affinity of the compound with the cationic linker, especially since its log D seems to be unchanged and thus cannot be the reason for a change in organ distribution. The authors should provide binding data to mCXCR4 or at least critically discuss this finding.

We thank the reviewers for pointing out an excellent missing piece in our manuscript. The relevant information has been added to the discussion:

In theory, the ammonium salt of N-methylpiperidine has a pKa of approximately 10.08, implying protonation of the piperidine-based linker at physiological pH, which should increase clearance rates. An explanation may be that the additional cationic charge results in a stronger interaction with erythrocytes, reducing the rate of extravasation of the radiotracer from the bloodstream into the tumor tissue; a similar effect is seen with [68Ga]Ga-/[177Lu]Lu-BL01, which is highly positively-charged.48 While a potential explanation for spleen and liver uptake is from mCXCR4-mediated interactions, the LY2510924 pharmacophore has a >1000-fold decrease in affinity to mCXCR4 as compared to hCXCR4, ruling out this explanation.49

Reviewer 2 Report

The Authors present a well-documented and appropriately designed study to demonstrate the effect of linker modification on their recently reported CXCR4 molecular imaging probe and potential theranostic agent, building on previous work exploiting this pharmacophore. 

The authors characterise a number of potential candidates both in vitro and in vivo, then select BL31 as a candidate based on comparative performance to their previously reported BL02 probe as well as the Pentixafor as the most currently used CXCR4 probe in the clinic. They conclude that the former has better performance based on higher tumour uptake and thus higher tumour to kidney ratio. Although the studies are well-designed and the data are presented well, I would suggest that a number of things could be usefully added to the discussion:

i) a single CXCR4 tumour line is tested: how does the receptor density of CXCR4 in Daudi xenografts compare to other lines used in the literature (U87.CXCR4) and to levels seen on patient tumour cells?

ii) although contrast of BL31 is demonstrated to be higher, the kidney uptake is still higher than Pentixafor. An extended discussion on why this would achieve a dose reduction in the clinic is warranted.

iii) the use of this probe as a theranostic agent is mentioned in the conclusion. A discussion of the caveats needed when extrapolating from a diagnostic to a therapeutic probe is therefore warranted, with reference to Pentixafor vs Pentixather.

iv) the lack of uptake in CXCR4-expressing organs in the mouse should be briefly noted and discussed 

Author Response

The Authors present a well-documented and appropriately designed study to demonstrate the effect of linker modification on their recently reported CXCR4 molecular imaging probe and potential theranostic agent, building on previous work exploiting this pharmacophore. 

The authors characterise a number of potential candidates both in vitro and in vivo, then select BL31 as a candidate based on comparative performance to their previously reported BL02 probe as well as the Pentixafor as the most currently used CXCR4 probe in the clinic. They conclude that the former has better performance based on higher tumour uptake and thus higher tumour to kidney ratio. Although the studies are well-designed and the data are presented well, I would suggest that a number of things could be usefully added to the discussion:

  1. i) a single CXCR4 tumour line is tested: how does the receptor density of CXCR4 in Daudi xenografts compare to other lines used in the literature (U87.CXCR4) and to levels seen on patient tumour cells?

We thank the reviewers for their comments. A thorough study that is cited in this manuscript is Kwon et al. Clin. Canc. Res. (2022), which thoroughly assesses several well-used hematological malignancy cell lines for CXCR4 expression. We find that [68Ga]Ga-BL02 able to effectively image tumor cell lines with even low expression of CXCR4, with excellent correlation of uptake and measured CXCR4 expression based on immunohistochemistry and flow cytometry. To address this comment, we have added the following sentence in the discussion:

“For consistency between previous studies, we used the Daudi Burkitt lymphoma xenograft model, which has shown to have robust CXCR4 and comparable CXCR4 expression as compared to other hematological malignancies.24

  1. ii) although contrast of BL31 is demonstrated to be higher, the kidney uptake is still higher than Pentixafor. An extended discussion on why this would achieve a dose reduction in the clinic is warranted.

We have added the relevant sentence addressing this comment in the body of the discussion:

“As such, the higher ratio of [68Ga]Ga-BL31 enables higher imaging contrast of lesions located proximally to the kidneys on PET imaging, though the absolute kidney uptake is greater.”

iii) the use of this probe as a theranostic agent is mentioned in the conclusion. A discussion of the caveats needed when extrapolating from a diagnostic to a therapeutic probe is therefore warranted, with reference to Pentixafor vs Pentixather.

We have added the relevant sentence addressing this comment in the body of the discussion:

“With respect to therapeutic applications, there is the concern that conjugation of another radiometal may abrogate the affinity of the radiotheranostic to the target receptor. This is shown with [68Ga]Ga-Pentixafor and [177Lu]Lu-Pentixather. We have previously demonstrated that [177Lu]Lu-BL02 shows no loss of performance with respect to both tumor uptake and non-target organ clearance; however, assessment of both in vitro binding affinity and in vivo performance is needed to definitively verify the applicability of [177Lu]Lu-BL31.”

  1. iv) the lack of uptake in CXCR4-expressing organs in the mouse should be briefly noted and discussed 

This has been addressed by the addition of this sentence in the discussion:

“While a potential explanation for spleen and liver uptake is from mCXCR4-mediated interactions, the LY2510924 pharmacophore has a >1000-fold decrease in affinity to mCXCR4 as compared to hCXCR4, ruling out this explanation.”

Reviewer 3 Report

The authors evaluated a series of radiotracers based on [68Ga]Ga-BL02, with modifications to its linker and metal chelator and disclosed an optimized radiopharmaceutical, [68Ga]Ga-BL31, containing a cysteic acid-based linker. I think this manuscript is worth publishing if the following questions get a good response.

1.In addition to cysteine, do other uncharged amino acids such as serine, threonine and tyrosine as linkers also lead to higher tumor-to-kidney ratios?

2.It would be more credible if the  explanations can be proved using related literatures in the last paragraph of the Discussion section.

3.Explain why there is better affinity and molar activity of [68Ga]Ga-BL31,compared with other agents in this study.

Author Response

The authors evaluated a series of radiotracers based on [68Ga]Ga-BL02, with modifications to its linker and metal chelator and disclosed an optimized radiopharmaceutical, [68Ga]Ga-BL31, containing a cysteic acid-based linker. I think this manuscript is worth publishing if the following questions get a good response.

1.In addition to cysteine, do other uncharged amino acids such as serine, threonine and tyrosine as linkers also lead to higher tumor-to-kidney ratios?

We thank the reviewers for the excellent suggestion. This study was primarily looking at the effects of charged linkers, with respect to modified anionic linkers, the addition of a cationic linker, and the use of a charged radiometal-chelate conjugate. Coupled with the constraints of word count of the manuscript, we elected not to explore the applications of hydrophilic linkers within the scope of this study.

2.It would be more credible if the  explanations can be proved using related literatures in the last paragraph of the Discussion section.

Additional references regarding the electrochemical properties and applications of sulfonic acids in drug design and as neurotransmitters have been added to the discussion.

3.Explain why there is better affinity and molar activity of [68Ga]Ga-BL31,compared with other agents in this study.

With respect to [68Ga]BL20 and [68Ga]BL30, only one radiochemical labeling run was performed and therefore, it is difficult to definitively say that [68Ga]Ga-BL31 showed higher molar activity when radiolabeled. With respect to [68Ga]Ga-BL02, there was no statistically significant difference between their molar activities. The separation of [68Ga]Ga-BL06 from non-labeled precursor was surprisingly difficult due to closer co-elution of peaks, resulting in reduced molar activity. This has been noted with the addition of the sentence:

“However, [68Ga]Ga-BL06 had closer co-elution of the radiolabeled peak and the unlabeled precursor, resulting in reduced molar activity.”

The reason we did not note the binding affinity was because it was not a statistically significant difference as compared to the other radiotracers.